# Measure Before You Look:
# Grounding Embeddings Through Manifold Metrics

**César Miguel Valdez Córdova**[1,2]    **Matthew Scicluna**[1,3]    **Shuang Ni**[1,3]
**Smita Krishnaswamy**[1,4] **Simon Gravel**[2]    **Guy Wolf**[1,3]
[1]Mila - Quebec AI Institute    [2]McGill University
[3]Université de Montréal    [4]Yale University
{cesar.valdez,matthew.scicluna,shuang.ni,wolfguy}@mila.quebec
simon.gravel@mcgill.ca
smita.krishnaswamy@yale.edu

## Abstract

Dimensionality reduction methods are routinely employed across scientific disciplines to make high dimensional data amenable to analysis. Despite their widespread use, we often lack tools to assess whether their resulting embeddings are faithful to the underlying manifold structure. Without a rigorous quantitative assessment of an embedding's structural properties, it is difficult to quantify their degree of preservation or distortion of the underlying manifold structure of the data. We introduce a complementary suite of geometric metrics to quantitatively audit embedding fidelity across neighborhood sizes: Tangent Space Approximation (TSA), Local Intrinsic Dimensionality (LID), and Participation Ratio (PR). We compare the dimensionality of each sample before and after embedding, where points that preserve similar values across transformations are deemed to be geometrically faithful and thus, representative of true manifold structure in the data.

Across synthetic and biological datasets, we show that these metrics expose distinct embedding failure modes: TSA is most sensitive to small-scale geometric distortions, LID captures heterogeneity in mixed-density regions, and PR diagnoses global variance structure. Finally, we demonstrate that applying Jacobian Frobenius penalties during autoencoder refinement of intermediate representations contracts tangent spaces, reduces disagreement between metrics, and improves alignment with intrinsic manifold geometry, as measured by rank correlations to original spaces. We motivate moving beyond visual heuristics and making principled, geometry-based choices to inform method selection, improve representations and motivate geometry-aware objectives for representation learning.

## 1 Introduction

Dimensionality reduction is used for visualization, clustering, and downstream learning across numerous domains ranging from single-cell transcriptomics to population genetics [1–3]. When projecting high-dimensional data into two or three dimensions, we implicitly assume that the embedding faithfully represents the underlying manifold [4]. This reliability is rarely questioned. However, embeddings that appear to be well-separated visually can distort local neighborhoods, collapse directions of variance, or warp tangent spaces in ways that aren't reliably visualizable in 2 or 3 dimensions [5, 6]. Common quantitative measures, such as explained variance, only provide partial guarantees. Embeddings can appear similar at a global scale but differ locally. This can lead to modeling inconsistencies. When dealing with high dimensional data, we assume it is generated from a lower-dimensional manifold. Modeling with representations that are inconsistent to this manifold geometrically but are otherwise visually consistent may result in misguided analyses.

Preprint.

In this work, we assess the structural fidelity of embedding spaces by integrating a complementary set of metrics spanning local and global scales, through neighborhood-based methods. We measure an embedding space's Local Intrinsic Dimensionality [7], Tangent Space Approximation [8], Participation Ratio [9], Trustworthiness and Continuity [10].

We contrast embeddings obtained from Principal Components Analysis (PCA) [11], which captures linear variation, against Archetypal Analysis (AA) [12] which emphasizes extreme points on a convex hull to understand how initial representation spaces influence subsequent embeddings. We then test whether embeddings can be actively refined to keep local geometry faithful across transformations via autoencoders, including variants penalizing the Frobenius norm of the Jacobian to smoothen latent geometry [13]. This introduces an explicit inductive bias: we trade off some global embedding fidelity, lost in reconstruction and through subsequent transformations, to obtain locally coherent latent spaces that better align with the intrinsic manifold structure of the embedding's source.

## 2 Methods

Our metric suite is designed to evaluate embedding spaces with a complementary set of geometric metrics that account for both local and global structure. This enables a comprehensive audit of embedding fidelity and overall manifold structure. Intrinsic dimensionality reflects the effective degrees of freedom in local data structure—preserving this indicates whether an embedding maintains the complexity of local manifold geometry rather than artificially inflating or collapsing it. Point-wise dimensionality estimates enable spatial diagnosis of where embeddings succeed or fail, unlike global metrics that can mask local failures.

For local metrics, we estimate the Tangent Space Approximation (TSA) by performing PCA locally within a point's neighborhood to infer the local intrinsic dimensionality required to explain a fixed fraction of variance. Local Intrinsic Dimensionality (LID) measures the effective number of degrees of freedom that are required to describe local structure, as determined by a maximum likelihood estimate of a point's neighbors' distances. Additionally, we employ Participation Ratio (PR) to estimate the effective number of variance directions, or eigenvalue spread, for a given local embedding patch. All of these are computed across neighborhood sizes $k \in \{5, 15, 25, 50, 100\}$. Trustworthiness and Continuity quantify how well local neighborhoods are preserved globally between the original ambient space and transformed embedded spaces. To ensure paired comparisons, we fix a random subset of $500$ evaluation points per dataset and reuse it consistently across all methods and seeds.

### 2.1 Datasets

We evaluate dimensionality reduction methods on two datasets, chosen to highlight distinct manifold geometries, across synthetic and real world settings. We use the Swiss Roll as a controlled benchmark: a synthetic 2 dimensional manifold projected into 200 dimensions, with known geodesic distances. Second, we assess high-dimensional population genomic data from the Human Genome Diversity Project (HGDP) [14] and Thousand Genomes Project (1KGP) [15] with mixed-mode distributions, known population admixture and multi-scale variation, providing a realistic test to assess embedding fidelity. See appendix B.1 and B.2 for further details about the Swiss Roll and HGDP+1KGP respectively.

### 2.2 Experimental Pipeline

Our goal is to evaluate how different initial representations and refinement steps affect the preservation of manifold structure across datasets. We designed a three-stage pipeline beginning with initial PCA or AA representations, followed by refined embeddings through both unregularized and regularized autoencoder reconstructions, and finally downstream embeddings evaluated with our comprehensive suite of manifold-aware metrics.

**Stage 1: Initial representations.** For each dataset, we first construct two 50-dimensional baseline representations:

- **$PCA_{50}$**: captures directions of maximal variance via principal components.
- **$AA_{50}$**: emphasizes extreme points on the convex hull as anchors, representing points as mixtures of these archetypes.

We use 50 dimensions as an intermediate representation to evaluate preservation at a realistic dimensionality commonly used in practice before final visualization, provide sufficient capacity for autoencoder refinement, and separate the effects of initial representation choice from final projection methods.

**Stage 2: Representation refinement with autoencoders.** We train two types of autoencoders to reconstruct the $PCA_{50}$ and $AA_{50}$ representations:

- **AE(PCA$_{50}$)** and **AE(AA$_{50}$)** Standard or "Vanilla" autoencoders trained by minimizing reconstruction error $L_{recon}$ on the baseline embeddings.
- **AE$_F$(PCA$_{50}$)** and **AE$_F$(AA$_{50}$)**. Autoencoders trained with an additional penalty on the Frobenius norm of the encoder Jacobian [13, 16]:

$$\mathcal{L} = \mathcal{L}_{recon} + \lambda_J \|J_f(x)\|_F^2$$

where $J_f(x)$ is the Jacobian of the encoder mapping $f$ with respect to its inputs, and $\lambda_J$ controls the strength of regularization.

This Frobenius penalty encourages the encoder to produce locally smooth, low-curvature latent spaces, making the refined embeddings less sensitive to noise and better aligned with intrinsic manifold geometry.

**Stage 3: Downstream embeddings.** We further reduce each of the representations from baseline and refinement stages into 2-dimensional spaces using two common projection methods: UMAP [17] and PHATE [18], where we sweep over their respective neighborhood parameters `n_neighbors` or `knn` $\in \{5, 15, 25, 50, 100\}$ to probe method and metric sensitivity to local or global structure. We then compute the Spearman's rank correlation from both the "refined" autoencoder and downstream embeddings with respect to their original representation spaces, AA or PCA. See appendix C for full architectural details.

## 3 Results

### 3.1 Our metric suite reliably estimates known intrinsic dimension on controlled data

We tested if these metrics captured the dimensionality of a dataset that is known to be 2-dimensional. The Local Intrinsic Dimensionality (LID), Tangent Space Approximation (TSA) and Participation Ratio (PR) reliably capture the intrinsic dimension of a Swiss Roll in 2 dimensions. The values for TSA diverge as the number of neighbors gets larger, starting at $k = 25$ with a more pronounced effect for the AA than the PCA initial embedding space. Participation Ratio stabilizes at $k = 25$ in AA, and right after in PCA. LID shows largely consistent values with a consistent divergence margin from 2 after $k = 5$. We found that regularizing the autoencoder with a Frobenius Jacobian penalty produces embeddings that are locally smoother and consistently lower-dimensional. Full definitions can be found in Appendix A.1. The full tables for PCA and AA baselines can be found in Appendix Table 2.

### 3.2 Improved local geometry propagates to downstream embeddings

To gauge consistency across embedding spaces, we use the Spearman rank correlation to quantify how well local neighborhood structure is preserved relative to the baseline embedding. See Figure 1 for LID, TSA, PR computed at $k \in \{5, 15, 25, 50, 100\}$ for all obtained embeddings. Each method's `n_neighbors` or `knn` parameter included in the figure matches the $k$ value at which it is evaluated. Frobenius-regularized autoencoders show a higher correlation to the original space for both the Swiss Roll and heterogenous HGDP+1KGP data. Exceptions are observed for PR and LID in the HGDP PCA cases, where a UMAP computed on the $50d$ baseline outperforms it.

## 4 Discussion

Our results establish two key findings: First, on controlled data (Swiss Roll), our metrics correctly identify known intrinsic dimensionality and reveal how different initial representations (PCA vs. AA) affect downstream geometric fidelity. Second, Frobenius regularization during autoencoder refinement

Table 1: Average metric performance of baseline methods on the Swiss Roll dataset across varying numbers of neighbors (k) for both PCA and AA ambient spaces. For PR, LID, and TSA, values closer to the intrinsic dimension of the dataset (2) are better.

| Method | Metric | Number of Neighbors (k) | | | | |
|---|---|---|---|---|---|---|
| | | 5 | 15 | 25 | 50 | 100 |
| PCA | Local Intrinsic Dim. (LID) | 3.33 | 2.04 | 1.82 | 1.70 | 1.85 |
| | Tangent Space Approx. (TSA) | 1.93 | 2.01 | 2.06 | 2.39 | 2.77 |
| | Participation Ratio (PR) | 1.50 | 1.68 | 1.76 | 1.95 | 2.21 |
| | Trustworthiness | 1.00 | 1.00 | 1.00 | 1.00 | 1.00 |
| | Continuity | 0.80 | 0.84 | 0.85 | 0.87 | 0.90 |
| AA | Local Intrinsic Dim. (LID) | 3.82 | 2.40 | 2.21 | 2.25 | 2.54 |
| | Tangent Space Approx. (TSA) | 2.14 | 2.77 | 3.11 | 3.81 | 4.96 |
| | Participation Ratio (PR) | 1.59 | 1.94 | 2.12 | 2.45 | 2.90 |
| | Trustworthiness | 1.00 | 1.00 | 0.99 | 0.99 | 0.99 |
| | Continuity | 0.47 | 0.55 | 0.60 | 0.68 | 0.74 |

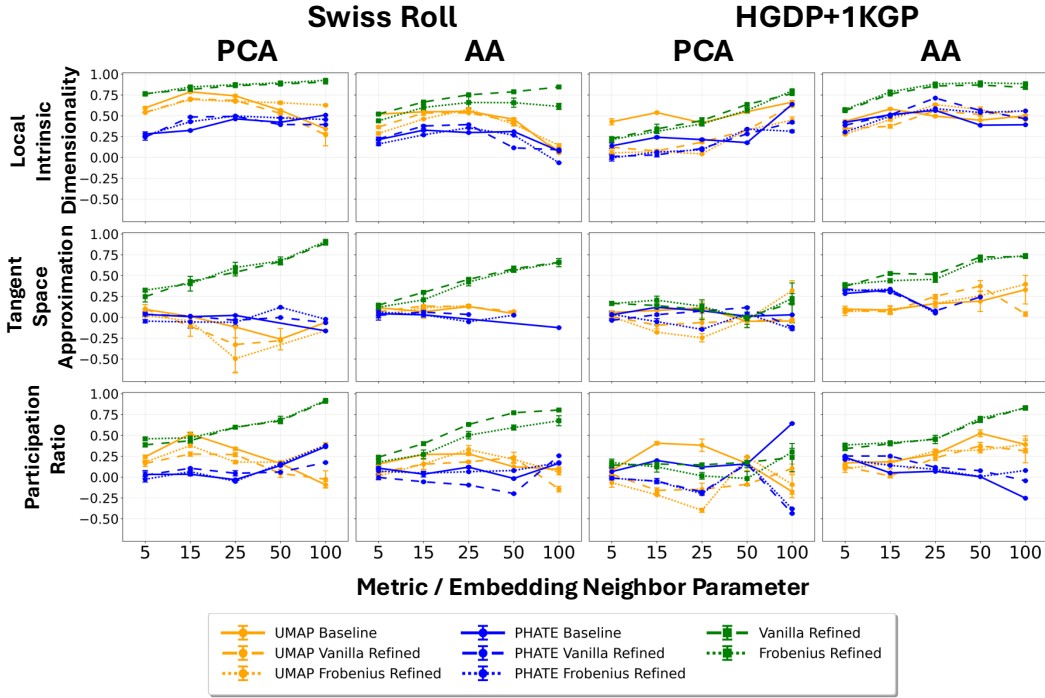

Figure 1: Correlations of average metric performance of reconstructions and downstream methods on the Swiss Roll and HGDP+1KGP datasets across varying numbers of neighbors $k$ against both of the original PCA and AA ambient spaces.

improves preservation of local geometric structure in intermediate representations, as evidenced by higher rank correlations in Figure 1. The practical implication is that *before* applying final 2D projection for visualization, practitioners can use these metrics to select intermediate representations that preserve more faithful geometry.

TSA excels for small, near-linear neighborhoods, making it well-suited to measure spaces with near linear variation. LID is strongest at medium values of $k$ in heterogeneous regions, and PR is informative at high values of $k$ but loses resolution when variance is concentrated along few directions.

This is especially apparent as noted by a divergence in metric agreement when measuring high dimensional AA spaces. As AA captures global extremal structure at the expense of local coherence, methods like TSA fail to capture local neighborhoods while LID and PR remain comparatively robust in successfully capturing intrinsic dimension. This is the center of our argument: internal geometrical coherence across embeddings isn't something that can be reliably seen with global or isolated metrics. Visual intuition builds on the former, therefore it is insufficient as a measure of embedding consistency; embeddings must be thoroughly audited before they're trusted to be representative of their data.

Embedding spaces can be actively steered via inductive biases like encoder-Jacobian Frobenius penalties. This geometric regularization contracts tangent spaces, reduces metric disagreement, and improves alignment between representations. The benefit starts to become apparent when the neighborhoods grow in heterogeneity, beyond tight local spaces at $k = 25$. Systematically quantifying these trade-offs establishes a foundation for principled, geometry-aware embedding refinement. We acknowledge an important limitation: methods like UMAP and PHATE project to 2D for visualization and cannot preserve local dimensionality beyond trivial cases. Our analysis focuses on comparing how different intermediate representations ($PCA_{50}$, $AA_{50}$, and their autoencoder refinements) maintain relative geometric consistency before final 2D projection. While all stages involve dimensional reduction, our metrics reveal which approaches better preserve local geometric structure through successive transformations. Our Frobenius penalty approach builds on contractive autoencoders [16], but differs in focus: while contractive autoencoders emphasize local invariance for robust feature learning, we explicitly audit geometric preservation across multiple scales to diagnose embedding fidelity.

This work establishes a diagnostic toolkit for auditing embeddings. Future work should develop clear protocols for using these metrics to guide embedding selection in practice, demonstrate tangible downstream benefits, extend validation to multiple synthetic manifolds with varying dimensionalities, and establish whether preserving local dimensionality in intermediate representations improves downstream tasks. These questions are essential for auditing representations produced by modern foundation models in a principled, geometry-aware way, allowing us to be intentional about what type of geometry we propagate through in our representations.

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

# A  Details of Measurements Used

## A.1  Common notation and metric definitions.

For each point $x_i$, let $N_k(x_i)$ be its $k$ nearest neighbors (excluding $x_i$ itself) in the ambient or embedded space. Form the local data matrix $X^{(i)} \in \mathbb{R}^{k \times D}$ by centering those neighbors, then compute its covariance

$$C^{(i)} = \frac{1}{k} \left( X^{(i)} \right)^\top X^{(i)}.$$

Let $\lambda_1^{(i)} \geq \lambda_2^{(i)} \geq \cdots \geq \lambda_D^{(i)}$ be the eigenvalues of $C^{(i)}$. We will build each metric from these $\{\lambda_j^{(i)}\}$.

**Participation Ratio (PR).**  The local participation ratio measures the effective number of dimensions utilized in a neighborhood. It is defined by

$$\mathrm{PR}(x_i) = \frac{\left( \sum_{j=1}^{D} \lambda_j^{(i)} \right)^2}{\sum_{j=1}^{D} \left( \lambda_j^{(i)} \right)^2}.$$

**Tangent Space Approximation (TSA).**  TSA estimates local intrinsic dimension by asking "How many eigenvalues are needed to capture at least a fraction $q$ of the total variance?" Concretely:

$$d_{\mathrm{TSA}}(x_i; q) = \min \left\{ d : \frac{\sum_{j=1}^{d} \lambda_j^{(i)}}{\sum_{j=1}^{D} \lambda_j^{(i)}} \geq q \right\},$$

where in our experiments $q = 0.95$.

**Local Intrinsic Dimensionality (LID).**  Unlike variance-based measures, LID estimates local dimensionality via a Maximum Likelihood framework based on neighbor distances. Let $d_j(x_i)$ be the distance to the $j$-th nearest neighbor of $x_i$, and let $d_k(x_i)$ be the distance to the $k$-th (i.e., furthest) neighbor in the set $N_k(x_i)$. The LID at point $x_i$ is the maximum likelihood estimate, given by:

$$\widehat{d}_{\mathrm{LID}}(x_i) = \left( -\frac{1}{k} \sum_{j=1}^{k} \log \frac{d_j(x_i)}{d_k(x_i)} \right)^{-1}.$$

**Continuity.**  Continuity measures the proportion of original high-dimensional neighbors preserved in the embedding:

$$\mathrm{Continuity}(k) = \frac{1}{n} \sum_{i=1}^{n} \frac{\left| N_k(x_i) \cap N_k'(x_i) \right|}{k},$$

where $N_k'(x_i)$ is the set of $k$-nearest neighbors of $x_i$ in the *embedded* space.

**Trustworthiness.**  Trustworthiness penalizes neighbors in the embedding that weren't true neighbors in the original space:

$$\mathrm{Trustworthiness}(k) = 1 - \frac{2}{n\,k\,(2n - 3k - 1)} \sum_{i=1}^{n} \sum_{j \in N_k'(x_i) \backslash N_k(x_i)} \left( r(i,j) - k \right),$$

where $r(i,j)$ is the rank of $j$ among the original high-dimensional neighbors of $i$.

## A.2  Rank-based Comparison

We use Spearman's rank correlation to assess whether embeddings preserve the *relative ordering* of local geometric properties across points. For datasets with heterogeneous structure, different regions naturally have different local dimensionalities—for example, on the Swiss roll, edge points may exhibit lower effective dimensionality than interior points due to boundary effects and sampling density. Rank correlation captures whether embeddings preserve these spatial patterns of geometric variation: a high correlation indicates the embedding maintains the relative complexity landscape

where regions that were locally more complex remain relatively more complex. This is more robust than absolute value comparisons, as it tolerates uniform scaling or shifts while detecting reorderings that indicate structural distortion.

Given corresponding metrics $M_i$ and $M'_i$ for each point $i$ in the original and embedded spaces respectively, Spearman's correlation $\rho$ is:

$$\rho = 1 - \frac{6 \sum_{i=1}^{n} (r_i - r'_i)^2}{n(n^2 - 1)}, \tag{1}$$

where $r_i$ and $r'_i$ denote the ranks of metric values $M_i$ and $M'_i$ respectively.

## B  Dataset details and pre-processing

### B.1  Swiss roll

We constructed a synthetic Swiss roll dataset to evaluate the capability of dimensionality reduction methods. The Swiss roll is designed to represent a two-dimensional plane $(y, t)$ that is smoothly embedded into a three-dimensional space $(x, y, z)$ through a spiral transformation. By default, the dataset consists of 100 distributions, each with 50 points, resulting in a total of 5000 samples. For each distribution, we sample random means for $t$ and $y$, and add Gaussian noise to introduce variability around these means. These $(y, t)$ coordinates encode the intrinsic geometry of the data manifold. We embed the 2D manifold into 3D space using:

$$x = t \cdot \cos t, \quad z = t \cdot \sin t,$$

while the y coordinate retains its noisy values. This transforms the flat plane into a 3D Swiss roll. After this transformation, additional Gaussian noise is added to all 3D coordinates to simulate observation or measurement noise.

To increase the complexity and evaluate robustness to ambient dimensionality, we further embed the dataset into a 200-dimensional space by applying a random orthogonal transformation. This projection preserves pairwise distances while eliminating any axis alignment, simulating high-dimensional real-world scenarios.

### B.2  HGDP + 1KGP

We combined HGDP and 1KGP whole-genome sequencing (WGS) 30X release [15]. To limit the genetics markers to a set of good quality and informative polymorphisms, we intersected this dataset positions contained on a largely used genotyping array (Illumina's GSAMD-24v3). This set of positions was then extracted from the 1000G dataset and additional filters were applied on this resulting intersection, including a maximum missing rate of 10% and minor allele frequency (MAF) 5% thresholds. We performed linkage disequilibrium (LD) pruning on the filtered 1000G dataset (plink v1.9 –indep-pairwise 50 5 0.5 parameters), followed by the exclusion of the human leukocyte antigen (HLA) region. The remaining missing data was imputed using ShapeIT5.

## C  Model architecture and tuning

### C.1  Vanilla Autoencoder (AE) Configuration

For baseline comparison, a standard autoencoder was trained for 150 epochs to reconstruct the 50-dimensional PCA and AA embeddings. The network's encoder mapped the 50-dimensional input to a 10-dimensional latent space through hidden layers of 32 and 16 neurons. The decoder then reconstructed the original 50-dimensional vector from this latent representation. The model was optimized using a standard Mean Squared Error (MSE) reconstruction loss, without any additional regularization on the latent space geometry. An early stopping validation loss was set with a patience of 10 epochs.

### C.2  Autoencoder with Frobenius Penalty ($AE_F$) Configuration

To investigate the effect of smoothing regularization, a second autoencoder ($AE_F$) was trained for 150 epochs on the 50-dimensional PCA embeddings. This model used an identical network architecture,

with an encoder compressing the input to a 10-dimensional bottleneck via hidden layers of 32 and 16 neurons. Its objective function was augmented with a Jacobian penalty. The model was trained to minimize a composite loss consisting of the MSE reconstruction error and a regularization term based on the squared Frobenius norm of the encoder's Jacobian, with the penalty weighted by a factor of $\lambda = 0.01$.

Table 2: Comparison of PCA and AA Baselines with Autoencoder Refinement and Frobenius Regularization. Metrics reported for varying neighborhood sizes ($k$). Note: values for LID $k = 15$ for both autoencoder cases were interpolated due to a mathematical error in computation.

| Method | Metric | k | Original | AE Refined | AE Frobenius |
|---|---|---|---|---|---|
| **AA** | Local Intrinsic Dim. | 5 | 3.82 | 3.63 | 3.75 |
| | | 15 | 2.20 | 2.16 | 2.21 |
| | | 25 | 2.20 | 2.16 | 2.03 |
| | | 50 | 2.19 | 2.16 | 1.98 |
| | | 100 | 2.41 | 2.40 | 2.15 |
| | Participation Ratio | 5 | 1.52 | 1.49 | 1.51 |
| | | 15 | 1.88 | 1.85 | 1.84 |
| | | 25 | 2.10 | 2.07 | 2.03 |
| | | 50 | 2.41 | 2.36 | 2.32 |
| | | 100 | 2.90 | 2.81 | 2.75 |
| | Tangent Space Approx. | 5 | 2.08 | 2.01 | 2.06 |
| | | 15 | 2.82 | 2.70 | 2.71 |
| | | 25 | 3.23 | 3.10 | 3.08 |
| | | 50 | 4.01 | 3.82 | 3.71 |
| | | 100 | 5.35 | 5.07 | 4.77 |
| | Trustworthiness | 5 | 0.99 | 0.99 | 0.99 |
| | | 15 | 0.98 | 0.97 | 0.98 |
| | | 25 | 0.97 | 0.96 | 0.97 |
| | | 50 | 0.96 | 0.95 | 0.96 |
| | | 100 | 0.94 | 0.94 | 0.94 |
| | Continuity | 5 | 0.47 | 0.48 | 0.46 |
| | | 15 | 0.54 | 0.55 | 0.54 |
| | | 25 | 0.58 | 0.59 | 0.57 |
| | | 50 | 0.64 | 0.64 | 0.63 |
| | | 100 | 0.68 | 0.69 | 0.67 |
| **PCA** | Local Intrinsic Dim. | 5 | 4.87 | 3.89 | 3.85 |
| | | 15 | 2.88 | 2.41 | 2.39 |
| | | 25 | 2.46 | 2.13 | 2.13 |
| | | 50 | 2.07 | 1.90 | 1.88 |
| | | 100 | 1.81 | 1.67 | 1.65 |
| | Participation Ratio | 5 | 1.65 | 1.57 | 1.55 |
| | | 15 | 2.03 | 1.77 | 1.76 |
| | | 25 | 2.06 | 1.78 | 1.79 |
| | | 50 | 2.11 | 1.92 | 1.88 |
| | | 100 | 2.53 | 1.92 | 1.88 |
| | Tangent Space Approx. | 5 | 2.17 | 2.07 | 2.05 |
| | | 15 | 2.79 | 2.29 | 2.28 |
| | | 25 | 3.04 | 2.30 | 2.31 |
| | | 50 | 3.40 | 2.49 | 2.47 |
| | | 100 | 3.53 | 2.69 | 2.58 |
| | Trustworthiness | 5 | 0.98 | 0.98 | 0.98 |
| | | 15 | 0.98 | 0.98 | 0.98 |
| | | 25 | 0.98 | 0.98 | 0.98 |
| | | 50 | 0.98 | 0.97 | 0.97 |
| | | 100 | 0.97 | 0.97 | 0.97 |
| | Continuity | 5 | 0.50 | 0.46 | 0.47 |
| | | 15 | 0.55 | 0.51 | 0.52 |
| | | 25 | 0.58 | 0.54 | 0.55 |
| | | 50 | 0.63 | 0.60 | 0.60 |
| | | 100 | 0.69 | 0.66 | 0.67 |

