# OpenReview forum: "Measure Before You Look: Grounding Embeddings Through Manifold Metrics"
_NeurIPS.cc/2025/Workshop/UniReps — UniReps2025_

### Official Review · Reviewer_xDGf · 2025-09-10
**Interesting idea tackling an important problem but with many methodological ambiguities and unclear results.**

**Confidence:** 3

**Review:**

The paper explores how much dimensionality reduction methods preserve the underlying manifold structure. They evaluate three geometric metrics and various neighborhood sizes.

Strengths:
- The idea is interesting and novel.
- The problem is important because dimensionality reduction of embeddings is used very often, but its faithfulness to the manifold structure needs to be explored.

Weaknesses:
- There are many methodological ambiguities, and the results are not well-presented. This probably hinders the quality of the conclusions. The following points should be understood mostly as something that was not clear to me and could perhaps be polished for the next version of the paper or used as an inspiration for future experiments.
	- It would be beneficial to include more details on the metrics used, mainly information that would help interpret the results: what values should we expect?
	- Is it enough to test a 2-dimensional manifold in a 200-dimensional space? I can imagine that the dimensionality reduction methods might reduce the dimensionality of the manifold as well, but there is not much space to reduce from 2 dimensions. What would happen, for example, with a 10-dimensional manifold?
	- Why do you use 50 dimensions in stage 1? In the motivation, it is mentioned that reductions to 2-3 dimensions are commonly used. In my opinion, the reduction 200->50 is not as huge as 200->2. How does this help evaluate the use cases mentioned in the motivation?
	- (perhaps connected to the previous one) Why is stage 2 necessary? Why not train an autoencoder directly from the original embeddings (so skipping stage 1)? Why is the reduction in stage 3 from the 50-dimensional embeddings and not the original ones? It seems to me that this dimensionality reduction is usually done at once, from the original to the target dimension. Why is it then beneficial to add more intermediate steps to this procedure?
	- I do not understand how the conclusions made in section 4 (Discussion) follow from the results presented in the previous section. I am not saying they are wrong, just that more understanding about what the values mean and how they are connected to the conclusions would help me as a reader to understand it better.
	- It is good that you also include a real dataset (HGDP+1KGP). However, some results for it are presented only in Figure 2, and I did not find any conclusions based on it. Why is it then useful to include this dataset in the analysis?
- Many of the references are incomplete, for example [1], [2], [4], [8], [12], [17].

**Score:**

2

**Topic Fit:**

3

---

### Official Review · Reviewer_VyEU · 2025-09-14
**Practical approach for figuring out geometric consistency of embeddings and with geometric metrics, preliminary in scope**

**Confidence:** 3

**Review:**

## Summary of the Work
The submission addresses a very important aspect of dimensionality reduction: the **faithfulness of embeddings to the underlying manifold** they represent.
To evaluate this, the authors propose a suite of complementary geometric metrics — **Tangent Space Approximation (TSA), Local Intrinsic Dimensionality (LID), and Participation Ratio (PR)** — to assess how faithfully embeddings preserve manifold structure at different neighborhood scales.
They also explore how applying **Jacobian penalties on the encoder** can improve local geometric alignment of embeddings.

---

## Strengths
- **Novelty**: Integrates existing metrics into a practical auditing suite with clear complementary roles.
- **Clarity of Motivation**: Provides a strong argument against relying solely on visual heuristics.
- **Empirical Validation**: Demonstrated on both controlled synthetic and complex genomics data.

---

## Weaknesses
- **Metric Interpretability**: Lacks guidance for practitioners when metrics diverge.
- **Comparisons**: Restricted to PCA, AA, and autoencoders. The evaluation would be stronger if extended to modern methods such as **contrastive learning** and other domains of **representation learning**.

---

## Suggestions
- Extend evaluation to **pretrained/self-supervised embeddings** (e.g., foundation models, contrastive methods).
- Provide a **practical decision framework** for interpreting conflicting metric outcomes.
- Discuss **scalability and computational efficiency** of metric computation.

**Score:**

3

**Topic Fit:**

2

---

### Official Review · Reviewer_sGHT · 2025-09-15
**Studying embedding faithfulness using a combination of different geometric measures**

**Confidence:** 4

**Review:**

**Summary:**
- The authors introduce a set of geometric measures to quantitatively assess the fidelity of low-dimensional embeddings obtained via various dimensionality reduction techniques. Across synthetic and biological datasets they show the proposed measures reveal different failure modes.
- They also propose obtaining "local geometry faithful" embeddings using autoencoders regularized by penalizing the Frobenius norm of the Jacobian to smoothen latent geometry (seemingly same as Rifai et al., 2011).

**Strengths:**
- The topic is of general interest and importance to the community.
- Exposition is clear and at the right level of abstraction.

**Feedback:**
- The proposed method with the Jacobian regularization seems to be the same as that of contractive autoencoders by Rifai et al., 2011. It is recommended that the authors look into it to see if it is indeed the same methodology, and if not, explain how they differ.
- Would it make more sense to have the AEs reconstruct the original data rather than the 50D embeddings? The current methodology implicitly assumes that the representations after step 1 still maintain the general geometry of the original space?
- The current set of experiments still doesn't leave us with any prescriptive advice or intuitions (other than maybe use regularized AEs going by Fig. 1); It would be useful to contrast different visualizations (e.g., in 2D) to see if any dimensionality reduction methods also apparently respect which geometric facets of the original data.

**References:**
- Rifai et al., 2011: Rifai, Salah, et al. "Contractive auto-encoders: Explicit invariance during feature extraction." Proceedings of the 28th International Conference on Machine Learning, 2011.

**Score:**

4

**Topic Fit:**

2

---

### Official Review · Reviewer_v1eb · 2025-09-15
**PCA and UMAP aren't enough (sometimes)**

**Confidence:** 4

**Review:**

This paper addresses an often-overlooked problem in data analysis: dimensionality reduction methods like PCA or UMAP can create visually appealing 2D plots that secretly distort the true underlying geometric structure of the original high-dimensional data. Relying on visual intuition alone can lead to flawed scientific conclusions.
To solve this, the authors propose a suite of quantitative geometric metrics to rigorously "audit" the faithfulness of an embedding. Instead of just looking at the final plot, their approach is to measure and compare the local geometry around data points before and after embedding.
The authors demonstrate on both synthetic and real-world genomics data that these metrics successfully identify different ways embeddings can fail. Furthermore, they show that embeddings can be actively improved. By training an autoencoder with a Jacobian Frobenius penalty, they create refined representations that are smoother and better preserve the true manifold structure of the data.



Strenghts:
- The validation on both a controlled synthetic dataset with a known ground truth (the Swiss Roll with dimension 2) and a complex, messy real-world dataset (human genomics) makes the findings highly credible and generalizable.

- Instead of a single score, the authors propose a suite of complementary metrics.


Weaknesses:
- The proposed metrics are neighborhood-based, which can be computationally intensive on very large datasets. The experiments were run on a fixed subset of 500 points, leaving the scalability of this auditing process to datasets with millions of samples as an open question.
- The paper doesn't explicitly show how a more "faithful" embedding leads to better outcomes on a downstream machine learning task like classification or clustering.

**Score:**

4

**Topic Fit:**

2